# Wastewater Sewage Sludge Management via Production of the Energy Crop Virginia Mallow

**Jona Šurić [1], Ivan Brandić [1], Anamarija Peter [1], Nikola Bilandžija [2], Josip Leto [3,*], Tomislav Karažija [4], Hrvoje Kutnjak [3], Milan Poljak [4] and Neven Voća [1]**

[1] Department of Agricultural Technology, Storage and Transport, Faculty of Agriculture, University of Zagreb, Svetošimunska Cesta 25, 10000 Zagreb, Croatia; jsuric@agr.hr (J.Š.); ibrandic@agr.hr (I.B.); apeter@agr.hr (A.P.); nvoca@agr.hr (N.V.)

[2] Department of Agricultural Engineering, Faculty of Agriculture, University of Zagreb, Svetošimunska Cesta 25, 10000 Zagreb, Croatia; nbilandzija@agr.hr

[3] Department of Field Crops, Forage and Grassland, Faculty of Agriculture, University of Zagreb, Svetošimunska Cesta 25, 10000 Zagreb, Croatia; hkutnjak@agr.hr

[4] Department of Plant Nutrition, Faculty of Agriculture, University of Zagreb, Svetošimunska Cesta 25, 10000 Zagreb, Croatia; tkarazija@agr.hr (T.K.); mpoljak@agr.hr (M.P.)

* Correspondence: jleto@agr.hr

**Abstract:** Wastewater treatment plants are facilities where wastewater is treated by technological processes. A byproduct of a wastewater treatment plant is sewage sludge, which can be both a good soil conditioner and a source of nutrients for the crops to which it is applied. Energy crops are non-food plants that can cleanse the soil of heavy metals through their ability to phytoremediate. The purpose of this study is to determine the effects of different amounts of sewage sludge on soil and plants. In the experiment Virginia mallow (*Sida hermaphrodita* L.) was used and the influence of stabilized sewage sludge in the amounts of 1.66, 3.32 and 6.64 t/ha dry matter on the energy composition and biomass yield was observed. The obtained results showed a yield of 8.85 t/ha at the maximum amount of sewage sludge used. Hemicellulose content was 20.20% in the application of 6.64 t/ha of sewage sludge and 19.70% in the control, while lignin content was 17.97% in the control and 16.77% in the maximum amount of sewage sludge. The heavy metals molybdenum and nickel did not differ significantly under the influence of larger amounts of sewage sludge, while manganese increased from 23.66 to 35.82 mg/kg.

**Keywords:** biomass production; yield; heavy metals; soil health; waste management

## 1. Introduction

Waste and its management are a symbol of inefficiency in modern society and an indicator of misdirected resources. Waste generation of any kind depletes natural resources and pollutes the environment, creating additional economic pressure on the strategic waste-management system [1–3]. At a time when energy sources are becoming very expensive, it is necessary to find ways to use energy resources from renewable sources, which waste undoubtedly is [4]. For this reason, the utilization of waste or by-products generated by any of the waste treatment processes is crucial in solving the problem of environmental protection and depletion of limited fossil reserves. One of the unavoidable by-products of wastewater treatment in a wastewater treatment plant is the sewage sludge [5]. The use of stabilized sewage sludge is only one of the possible solutions for the replacement of mineral fertilizer, which is currently reaching its maximum price. The originality of this paper lies in the amount used, which is three times higher than the amount allowed in Croatia for agricultural production. The increasing amounts of sewage sludge deposited in landfills show that its disposal or incineration is not an adequate solution for sewage sludge management and that it must be used for other purposes [6]. In

the European Union, there is no clear legal or practical definition for the management of municipal sewage sludge. Thus, it is not surprising that each member state handles all of its sewage sludge in its own way, which further complicates the sewage sludge problem [7]. The problem of sewage sludge is addressed in the Directive (1999/31/EC) [8] of the European Parliament and the Council on waste, which regulates the recycling of waste, including sewage sludge from secondary treatment plants. One of the main objectives of the directive is to maximize the utilization of the nutrients contained in the sludge in agriculture, while complying with all sanitary, chemical, and environmental safety regulations. Due to the organic substances contained in sewage sludge, it can be considered a good substrate for fertilization or an aid for the rehabilitation of degraded soils. Such reuse of sewage sludge is in line with the European Commission's main concept "reduce, recover, recycle" [9].

Aerobic digestion is only one of the technological processing options that helps to reduce the volume of sewage sludge and also reduce putrefaction, which significantly affects the occurrence of unpleasant odors [10,11]. The occurrence of unpleasant odors, in addition to the health risk in the form of excessive concentration, leads to dissatisfaction of the local population, as there is an aversion to the use of sewage sludge [12].

Shredding and the introduction of pretreatments (high pressure, microwave ultrasound) contribute to the decomposition of organic matter, but also to the destruction of certain microorganisms, which becomes a questionable main function for the soil in the context of soil conditioners [13,14].

The aerobic stabilization that occurs during anaerobic digestion contributes to the production of high yields of methane-rich biogas [15].

On the other hand, sewage sludge treatment is an expensive and demanding process that includes various biological, chemical, and thermal treatments aimed at removing pathogenic organisms and reducing volume [16]. The use of sewage sludge for soil conditioner is not an unknown form in countries around the world. This is confirmed by the fact that 40% of sewage sludge produced in Europe is applied to agricultural soils, and the results show better soil structure and properties, as well as higher yield [9,17]. Such use of sewage sludge seems to be a reasonable and effective solution for disposal, but only in the case of neutralization of complex or potentially toxic properties of sewage sludge [18]. Recycling nutrients from renewable sources, such as sewage sludge, through agricultural applications could be an important step toward a circular economy that may lead to a reduced need for inorganic fertilizers [19].

Slightly less than 50% of sewage sludge is organic matter available for crops. On the other hand, it is a potential hazard to humans and the environment due to the presence of heavy metals and organic pollutants, leading to discussions on new techniques for its processing and disposal through the recovery of nutrients [20].

When sewage sludge is applied to agricultural land, it is important to know that it is a stabilized sludge in which pathogenic microorganisms are killed so as not to pose a threat to the environment [21]. In addition, the recycling of nutrients from organic waste and the circulation of phytonutrients allow agricultural production at a lower cost [22–24].

Stabilized and treated sewage sludge can meet the nutrient needs of energy crops and be an adequate substitute for mineral fertilizers [21]. According to [25], energy crops are plants grown on degraded soils for the purpose of biomass production. The main reason for cultivation is to produce renewable energy, substitute fossil energy resources, and make an important contribution to environmental protection, as well as to end competition between energy and forage crops. Energy yield and energy efficiency are the most important criteria in evaluating biomass-based energy crops and renewable energy chains [26,27]. One such energy crop is Virginia mallow (*Sida hermaphrodita* L.). Virginia mallow is a perennial herbaceous plant in the mallow family (lat. *Malvaceae*) and is native to North America, where it grows near wetlands, floodplains, and rivers [28,29]. Once planted, Virginia mallow can be productive for at least 25 years, with some authors suggesting that it

can remain very productive for even longer [30]. The interest in Virginia mallow production is related to its minimal agronomic requirements, promotion of biodiversity, and high resistance to diseases and pests [31].

Virginia mallow has the ability to store carbon in a well-developed and branched root system, which is a great advantage when grown on marginal soils because it allows efficient use of nutrients and water [32,33].

Virginia mallow can adapt to very low temperatures, which means it is suitable for continental landscapes [34]. Yields are significantly reduced when there is no rain. According to [35], plants in humid continental climates require a minimum annual rainfall of about 500–600 mm. In addition to moisture, pH also affects biomass yield. Yield can be significantly reduced in acidic soils, as shown by the research of [36], who increased pH from 4.3 to 5.6 by liming, resulting in a 50% increase in yield.

Here we show the influence of municipal sewage sludge on energy properties and biomass yield of Virginia mallow. In addition to yield components, the objective was to analyze whether increasing the dose of applied municipal sewage sludge from wastewater affected the chemical properties of the soil and whether there were changes in soil fertility and heavy metals settling.

## 2. Materials and Methods

### 2.1. Practical Field

The practical field with Virginia mallow was based on experimental field Maksimir (elevation 123 m, 45°49′48″ N 16°01′19″ E) at the Faculty of Agriculture, University of Zagreb. In 2017, the seedlings were planted with a spacing of 0.75 m within and between rows, according to the split-plot principle in three replicates. The base plot is 8.44 m$^2$, while the subplot is 4.22 m$^2$. The experiment was laid out according to the split-plot scheme, and each treatment was applied in 3 replicates (0 t/ha (control), 1.66, 3.32, and 6.64 t dry matter/ha). Stabilized sewage sludge was applied once to agricultural soils where Virginia mallow was planted. The sewage sludge was applied once and mixed with the topsoil layer in the experiment in March 2019, while harvesting occurred one year later, in March 2020. The monthly average temperature and precipitation values in the Virginia mallow experimental field during the growing season are shown in Figure 1.

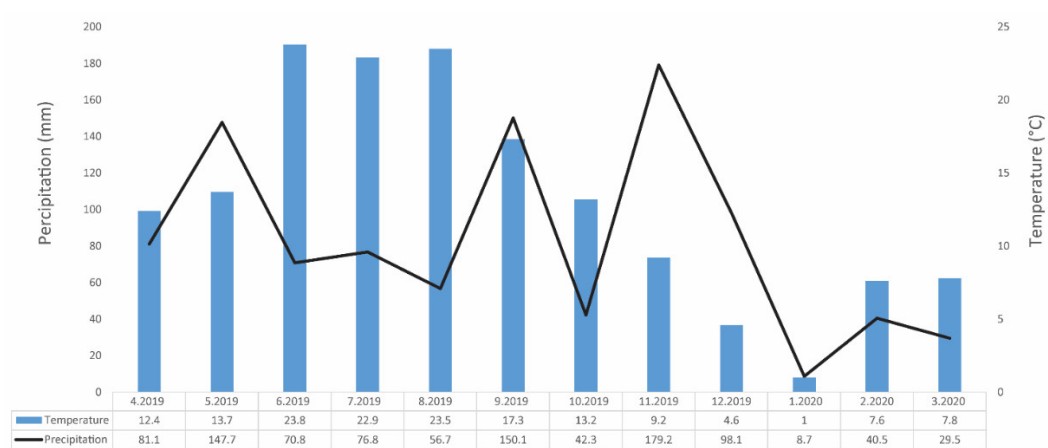

**Figure 1.** The average monthly precipitation and temperature at experimental field of Virginia mallow (Maksimir, elevation 123 m, 45°49′48″ N 16°01′19″ E).

### 2.2. Stabilized Sewage Sludge and Soil Characteristics

Sampling of the sewage sludge was carried out before application to the soil in order to determine the characteristics of the soil in accordance with the Croatian legislation for agricultural application of municipal sewage sludge. The municipal sewage sludge used

for the experiment was collected from a sewage treatment plant in Zagreb and its surroundings and was subjected to laboratory analysis before being applied to the soil. The plant is located in the eastern part of the capital and has a capacity of 1.5 million PE (population equivalent). Approximately 50,000 tons of sewage sludge are produced annually. Table 1 shows the chemical properties of the municipal sludge (dry weight) used in the experiment. The pH was determined with a glass electrode using a ground–water suspension 1:10 (*w/v*), electrical conductivity was determined in a conductometer (soil/water ratio 1:10) at 25 °C.

**Table 1.** Chemical properties of sewage sludge and soil before application of different amounts of sewage sludge.

| Unit | | mS/cm | | | % | | | |
|---|---|---|---|---|---|---|---|---|
| Items | pH | E.C. | C Organic | N Total | $P_2O_5$ Total | $K_2O$ Total | Ca Total | Mg Total |
| Sewage sludge | 12.44 | 7.51 | 25.91 | 3.72 | 2.53 | 0.46 | 13.47 | 0.76 |
| Soil | 6.83 | 0.035 | - | 0.13 | - | - | - | - |

E.C.—electrical conductivity, C—carbon, N—nitrogen, $P_2O_5$—phosphor total, $K_2O$—potassium total, Ca—calcium, Mg—magnesium.

Soil samples (0–30 cm) and sludge samples were collected before the sewage sludge was applied to determine the content of heavy metals. Table 2 shows whether the determined heavy metal values in soil and sewage sludge are in accordance with the values prescribed in the Croatian legislation for the use of municipal sewage sludge in agriculture [37].

**Table 2.** Sewage sludge used in accordance with the values prescribed by Croatian legislation.

| | | | | Permitted Total Heavy Metals Content [32] | |
|---|---|---|---|---|---|
| Items | Unit | Sewage Sludge | Soil | Sewage Sludge | Soil (pH mKCl > 6.5) |
| Cd | mg/kg | 0.14 | <1.0 | 5 | 1.5 |
| Cu | mg/kg | 258 | 28.60 | 600 | 100 |
| Ni | mg/kg | 22.8 | 31 | 80 | 70 |
| Pb | mg/kg | 42.7 | 25.4 | 500 | 100 |
| Zn | mg/kg | 543 | 76.23 | 2000 | 200 |
| Cr | mg/kg | 51.8 | 85.8 | 500 | 100 |

Cd—cadmium, Cu—copper, Ni—nickel, Pb—lead, Zn—zinc, Cr—chromium.

The content of heavy metals in the sludge was measured during the extraction of imperial water using the AAS graphite and hydride technique (SOLAR AA spectrometer M series, Thermo Fisher Scientific, Waltham, Massachusetts, SAD, using Graphite Furnace and Cold Vapor System).

For soil analysis, 50 individual samples were taken to obtain an average value. Soil core samples (0–30 cm) were collected from each postharvest Virginia mallow fertilizer treatment to determine changes in soil properties. "Soil organic carbon was determined using the $K_2Cr_2O_7$ method of external heating according to modified Tjurin method [38]. Soil pH was determined using a pH meter (soil to water ratio (5:1) [39]." Total nitrogen (N) in the soil was determined by the Kjeldahl method. Available phosphorus (P) and potassium (K) were measured using AL method; extraction with ammonium lactate-acetic acid at a ratio of 1:20 (*m/v*), after which the phosphorus concentration was determined by graphite and the potassium by flame analysis on spectrometer. Total calcium was determined by the volumetric method [40], and available magnesium by the standard method for soil analysis (14-3,1,1: 1982).

*2.3. Yield Components and Energy Biomass Characterization*

To determine the characteristics of the plant, samples were taken in March. Plant height was measured in meters from the bottom to the top of the flower on ten randomly selected plants. Dry matter yield was determined by cutting the plants at a height of few cm above the ground. The pod was then weighed and removed and placed in a fan dryer (model 30-1060, Memmert, Schwabach, Germany) at 60 °C for 48 h. The dried subsample was weighed again to determine the tonnage of dry matter per hectare [41].

About two kilograms of the dried sample, the above-ground portion of the biomass, was used to determine the energetic properties. The dry sample was first coarsely ground in a mixer (Retsch Grindomix GM 300, Haan, Germany) and then processed in a laboratory mill (IKA Analysentechnik GmbH, Staufen, Germany) [42] and sieved on a sieve shaker (Retsch AS200, Germany). The samples were then analyzed in the laboratory of the Department of Technology, Storage and Transport, Faculty of Agriculture, University of Zagreb. For laboratory analyses, 150 g of the dry sample with a particle size of 250 μm–1000 μm was taken and further analyzed according to standard methods.

Coke content [43] and ash [44] were determined using a muffle furnace (Nabertherm GmbH, Nabertherm Controller B170, Lilienthal, Germany). Content of volatile-matter and fixed carbon were calculated mathematically by difference.

Nitrogen, hydrogen, carbon [45], and sulfur [46] contents were determined by dry combustion in a Vario Macro CHNS analyzer (Elementar Analysensysteme GmbH, Langenselbold, Germany). The oxygen content was determined mathematically from the difference.

Content of macroelements (Ca, K, Mg) [47] and heavy metals (Cu, Mn, Zn, Mo, Ni, Cr, Pb) [48] in biomass was determined with prior closed digestion of a sample in a closed-digestion microwave system (Milestone Ethos D, Sorisole, Italy), using an atomic absorption spectrometer (Perkin Elmer AAnalyst 400, Waltham, MA, USA).

Cellulose content, hemicellulose content (NREL/TP-510-42623) and lignin content (NREL/TP-510-42618) was determined under laboratory conditions using ANKOM 2000 Fiber Analyzer (Macedon, New York, NY, USA).

Higher calorific value [49] was determined at IKA C200 Analysentechnik GmbH Heitersheim (Staufen, Germany), and lower calorific value was calculated mathematically using the ISO method.

After the laboratory analyses, statistical processing and interpretation of the data was performed using the program Statistica package version 10 PL. Means and standard deviation were tested using Tukey's HSD test with significance level ($p < 0.05$).

*2.4. Statistical Analyses*

The data were processed statistically using the software package STATISTICA 10.0 (StatSoft Inc., Tulsa, OK, USA). All analyses were conducted in three replicates. The obtained results are expressed as the mean value with standard deviation (SD). Analysis of variance (ANOVA) with Tukey's HSD post hoc test ($p \leq 0.05$) for comparison of the sample means were used to explore the variation of the observed parameters. All observed samples were checked for variance equality (using Levene's test) and normal distribution (using Shapiro–Wilk's test).

## 3. Results and Discussion

When the basic soil chemical parameters were analyzed, it was found that there were statistically no significant changes between the control and treated plots, as shown in Table 3. Phosphorus, potassium, and pH decreased under the treated conditions with increasing sludge application compared to the control treatment, while ammonia and nitrate nitrogen increased with increasing amounts of sludge applied. Total nitrogen remained the same regardless of treatment with different amounts of sewage sludge.

**Table 3.** Basic soil chemical properties after application of different amounts of sewage sludge and cultivation with Virginia mallow.

| Unit | pH | | % | | mg/100 g | | mg/100 g | |
|---|---|---|---|---|---|---|---|---|
| Sludge Treatment | $H_2O$ | mKCl | hum | N | $NO_3^-$ | $NH_4^+$ | $P_2O_5$ | $K_2O$ |
| T1 | 7.74 [a] | 6.68 [a] | 1.72 [a] | 0.14 | 0.59 [a] | 0.87 [a] | 25.83 [a] | 13.50 [a] |
| T2 | 7.82 [a] | 6.69 [a] | 1.71 [a] | 0.14 | 0.62 [a] | 0.83 [a] | 24.80 [a] | 12.73 [a] |
| T3 | 7.68 [a] | 6.55 [a] | 1.74 [a] | 0.14 | 0.63 [a] | 1.04 [a] | 23.20 [a] | 12.70 [a] |
| T4 | 7.59 [a] | 6.47 [a] | 1.69 [a] | 0.14 | 0.75 [a] | 0.95 [a] | 24.03 [a] | 12.23 [a] |
| Average | 7.71 | 6.60 | 1.72 | 0.14 | 0.65 | 0.92 | 24.47 | 12.79 |
| St.dv. | 0.10 | 0.11 | 0.02 | - | 0.07 | 0.09 | 1.12 | 0.53 |
| Significance | ns | ns | ns | - | ns | ns | ns | ns |

T1—(control), T2—1.66 t/ha, T3—3.32 t/ha, T4—6.64 t/ha. Values are the mean SD of three replicates. Different letters in each row indicate significant difference at $p \leq 0.05$ by Tukey's HSD test. Values without letters are not significantly different at $p \leq 0.05$ by Tukey's HSD test. Significance: ns, non-significant.

In previous studies where municipal sludge was applied to the soil, there was an increase in nitrogen, phosphorus, and potassium when the sludge dose was increased. This result is due to the quantities applied. Namely, previous studies used between 10 and 60 t/ha DM of sewage sludge, which is up to 10 times higher than the amounts used in this study [50].

Principal component analysis (PCA) was used to examine the relationship between the observed soil samples and the elements concentration in these samples, as well as to group the samples according to similarity according to the observed parameters.

PCA analysis has been successfully applied to classify and separate different samples at the factor space (based on measured concentration of heavy metals). Similarity recognition techniques based on experimental values have been applied to separate samples at the factor space.

The results of the measured parameters (descriptors) are graphically presented on the PCA graph for soil samples in Figure 2, using the first two main components, which were obtained from PCA analysis.

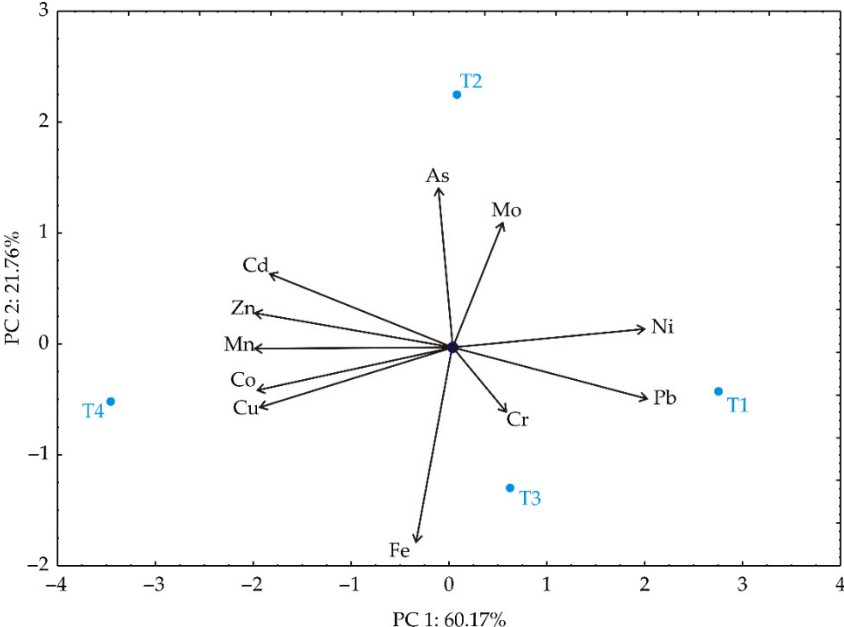

**Figure 2.** Biplot graph of soil samples based on heavy metals concentration. T1—(control), T2—1.66 t/ha, T3—3.32 t/ha, T4—6.64 t/ha.

Figure 2 shows that a successful separation of four samples (T1, T2, T3, and T4) was performed. Qualitative results for this analysis show that the first three main components together represent 81.93% of the total variance, which can be considered sufficient to present the whole set of experimental data.

The content of Ni (13.7% of the total variance, based on correlation) and Pb (14.1%), had the most significant positive impact on the calculation of the first principal component (PC1), while the content of Cd (12.7%), Zn (14.5%), Mn (14.5%), Co (14.1%) and Cu (13.8%) showed the negative influence of PC1 calculation. The most significant positive influences on the calculation of the second principal coordinate (PC2) were achieved through the concentrations of As (26.1% of the total variance, based on correlation) and Mo (16.1%), while the most pronounced negative influence on PC2 was recorded for the content of Fe (38.3%).

The values of the observed variables (contents of heavy metals) in different samples can be estimated from Figure 2.

According to Figure 2, sample T1 has higher contents of Ni and Pb and is located on the right side of the graphic. Sample T4 was reached in Cd, Zn, Mn, Co, and Cu concentrations. The increased concentration of As and Mo was observed in the T2 sample, while the augmented concentration of Fe was noticed in sample T3. According to the Tukey's HSD test, none of the heavy metals concentrations were significantly different between samples (T1–T4).

According to [51], individual plants can reach a height of up to 3 m during this period, which is confirmed by this study in Table 4. When sewage sludge was applied at the highest concentration of 6.64 t/ha, plants reached a height of more than 3 m, indicating that sewage sludge has favorable properties for plant growth and can adequately replace the previous use of standard fertilizers.

**Table 4.** Yield results and yield components of Virginia mallow after application of different amount of sewage sludge.

| Unit | m | t/ha | % |
|---|---|---|---|
| Sludge treatment | Plant height | Dry matter yield | Dry matter |
| T1 | 3.12 [a] | 6.53 [a] | 76.87 [a] |
| T2 | 2.99 [a] | 5.72 [a] | 80.54 [a] |
| T3 | 3.01 [a] | 8.93 [a] | 80.61 [a] |
| T4 | 3.28 [a] | 8.85 [a] | 80.35 [a] |
| Average | 3.1 | 7.51 | 79.59 |
| St.dv. | 0.13 | 1.63 | 1.82 |
| Significance | ns | ns | ns |

T1—(control), T2—1.66 t/ha, T3—3.32 t/ha, T4—6.64t/ha. Values are mean ± SD of three replicates. Different letters in each row indicate significant difference at $p \leq 0.05$ by Tukey's HSD test. Values without letters are not significantly different.

The observed tendency that yield increases with a higher concentration of applied sewage sludge is in agreement with the studies of [21]. Namely, they show a significant difference in biomass yield when 10 t/ha and 60 t/ha were used, with yield increasing from 11 to 15 t/ha. The results indicate that sewage sludge can make a positive contribution to increasing biomass yield, but this increase is only significant when the minimum amount of sewage sludge applied is at least 10 t/ha. In study [52], yields in the range of 9–11 t/ha were obtained when sewage sludge was applied, which is slightly higher than the results obtained in this study.

Dry-matter yield and biomass quality depend largely on harvest date [53]. In fact, harvesting in winter results in significantly lower moisture and ash content, which improves biomass quality for direct combustion. A lower percentage of moisture in the biomass results when harvesting is delayed until spring compared to fall. In this way, natural

drying is achieved in the field, which reduces the yield, but also the amount of water in the plant, which greatly facilitates the storage of the harvested biomass [54]. The loss of moisture inside the plants also leads to the release of leaves, which ultimately results in lower ash content, as most of the minerals remain inside the leaves during combustion in the ash [55]. The moisture content of Virginia mallow biomass decreases from about 40% in November to about 20% in spring, which allows direct pelleting after harvest without pretreatment [56].

For direct combustion, the approximate analysis of biomass is of particular importance, while it is less pronounced for liquid biofuels and almost negligible for gaseous biomass. Table 5 shows that the volatile-matter content changed only slightly after the application of sewage. The increase in volatile-matter content was accompanied by an increase in oxygen content, which was also observed in the table with an increase in sewage sludge volume. The term volatile matter refers to constituents in a fuel that are released by heating at very high temperatures. Due to the high volatile content, most of the biomass vaporizes before a homogeneous combustion phase occurs, while the remaining charred residue enters a heterogeneous combustion reaction [57]. Vaporization during biomass heating occurs as long as there are volatile compounds in the biomass.

**Table 5.** Proximate analysis of Virginia mallow biomass after application of different amounts of sewage sludge.

| Unit | % | | | | MJ/kg | |
|---|---|---|---|---|---|---|
| Sludge Treatment | Ash | Coke | Fixed Carbon | Volatile Matters | HHV | LHV |
| T1 | 2.65 [a] | 10.33 [a] | 7.89 [a] | 81.92 [ab] | 17.29 [a] | 15.95 [a] |
| T2 | 3.00 [a] | 10.74 [a] | 7.93 [a] | 82.87 [b] | 17.12 [a] | 15.76 [a] |
| T3 | 2.98 [a] | 10.83 [a] | 8.04 [a] | 82.38 [ab] | 17.36 [a] | 16.02 [a] |
| T4 | 2.76 [a] | 11.19 [a] | 8.65 [a] | 81.23 [a] | 17.26 [a] | 15.91 [a] |
| Average | 2.85 | 10.77 | 8.13 | 82.10 | 17.26 | 15.91 |
| St.dv. | 0.39 | 0.64 | 0.71 | 1.26 | 0.28 | 0.27 |
| Significance | ns | ns | ns | *** | ns | ns |

T1—(control), T2—1.66 t/ha, T3—3.32 t/ha, T4—6.64 t/ha. Values are the mean SD of three replicates. Different letters in each row indicate significant difference at $p \leq 0.05$ by Tukey's HSD test. Values without letters are not significantly different at $p \leq 0.05$ by Tukey's HSD test. Significance: *** $p < 0.001$; ns, non-significant.

One of the most important properties of biomass is its ash content. Due to its complex content, alternating organic and inorganic components, it can cause serious problems in combustion boilers and even lead to system plugging [58]. Since ash-containing feedstocks have no calorific value, it is desirable to use them in the lowest possible concentrations. In this study, the ash content remained the same regardless of the amount of sludge applied. This was not the case in other studies where the ash content ranged from 2 to 5% [59,60].

Solid carbon, together with ash, forms a solid residue after combustion, i.e., the release of volatile substances. Fixed carbon conditions the production of biochar and burns as a solid component, which means that it is desirable to have as much of it as possible due to its positive effect on combustion [61]. The different amounts of stabilized sewage sludge used in this study did not significantly affect the fixed carbon content. However, the growth trend showing the potential of using sewage sludge as fertilizer for energy crops should not be overlooked.

The growth trend with increase in sewage sludge quantity is also observed for coke, where the biomass with the highest sewage sludge dose contained the highest percentage of coke, 11.19%. Coke increases the quality of the fuel and remains as a by-product after dry distillation. It is desirable to have it in as large quantities as possible. Thus, the increase

recorded in this study shows that the use of sewage sludge favors an increase in coke content in Virginia mallow.

Another important indicator of the quality of the fuel is its calorific value. This value expresses the amount of energy released in the complete combustion of one unit mass of the fuel, cooling the gases to 25 °C and expelling water as condensate [54]. The water content in the fuel and the calorific value are inversely proportional, which means that the calorific value decreases as the amount of water increases [62]. Table 5 shows the higher (HHV) and lower (LHV) heating value of Virginia mallow grown in four stages of sewage sludge fertilization. The lower heating value shows the energy content of the fuel without the heat of condensation of the water vapor contained in the exhaust gases. In the case of biomass, the difference between the higher and lower heating values represents the loss due to flue gases in conventional boilers that cannot use the heat energy extracted from the water vapor [54,63].

Combustible substances are substances that, in the presence of atmospheric oxygen, are brought to their ignition temperature and pass into gaseous compounds and noncombustible residues, forming flames or embers. This process primarily involves the aforementioned oxygen, which does not burn but allows combustion to occur, and other elemental constituents such as sulfur (S), hydrogen (H), nitrogen (N), and carbon (C). The main element of biomass is carbon, which does not occur freely in biomass but is found in organic compounds along with other elemental compounds. When burned, carbon binds to oxygen and produces a large amount of heat energy. The average carbon content of Virgin mallow biomass is 51.50%, which is higher than the results of other literature where the carbon content does not exceed 50% [64–66]. Interestingly, in this study, the carbon content decreased with the increase in sewage sludge volume, so that 51.44% of carbon was obtained at applied 6.64 t/ha sewage sludge, while the carbon content at applied 1.66 t/ha sewage sludge was 51.72%. Due to the large and important role that hydrogen plays in biofuels, it is important to have it in as high a percentage as possible. In this experiment, when the maximum dose of sewage sludge was applied to Virginia mallow, the amount of hydrogen increased significantly which are listed in Table 6. The increase obtained is the only significant increase observed in the percentage of macroelements compared to the control values. Since the content of carbon and hydrogen increased with the increasing amount of applied sewage sludge, it can be concluded that sewage sludge certainly has a positive effect on the two most important elements of biomass for direct combustion.

**Table 6.** Elemental analysis of Virginia mallow biomass after application of different amounts of sewage sludge.

| Unit | % | | | | |
|---|---|---|---|---|---|
| Sludge Treatment | C | H | N | S | O |
| T1 | 51.72 [a] | 6.16 [ab] | 0.18 [a] | 0.06 [a] | 41.88 [a] |
| T2 | 51.57 [a] | 6.15 [b] | 0.18 [a] | 0.05 [a] | 41.96 [a] |
| T3 | 51.28 [a] | 6.18 [a] | 0.17 [a] | 0.08 [a] | 42.32 [a] |
| T4 | 51.44 [a] | 6.23 [ab] | 0.21 [a] | 0.06 [a] | 42.11 [a] |
| Average | 51.50 | 6.18 | 0.19 | 0.06 | 42.07 |
| St.dv | 0.35 | 0.22 | 0.04 | 0.02 | 0.37 |
| Significance | ns | *** | ns | ns | ns |

T1—(control), T2—1.66 t/ha, T3—3.32 t/ha, T4—6.64 t/ha. Values are the mean SD of three replicates. Different letters in each row indicate significant difference at $p \leq 0.05$ by Tukey's HSD test. Values without letters are not significantly different at $p \leq 0.05$ by Tukey's HSD test. Significance: *** $p < 0.001$; ns, non-significant.

Nitrogen does not participate in the combustion process, does not generate heat, and therefore reduces the calorific value of the fuel [63]. However, what works, and why biomass is better the lower the nitrogen content, is that it reduces the calorific value of the fuel by binding to elements that are actively involved in combustion, preventing their activity. It is also considered to be harmful to the environment, as it can form undesirable nitrogen oxides (NOx) during combustion [67]. The mean value for nitrogen in this study was 0.19%, which is slightly lower than the values obtained in previous studies, but in the case of nitrogen, this is a positive result. For example, [66] measured 1.5% nitrogen, [64] and [68] each measured 0.5%, while [69] determined 0.34%.

Nitrogen is not the only one that can be involved in combustion pollution; there is also sulfur, which forms sulfur oxide (SOx) during combustion. It can be flammable and then bind to the organic part or to metals, or it can be nonflammable if it remains in the ash as calcium sulfate. It is present in the biomass in very small quantities and therefore does not significantly affect the quality of biomass combustion. The use of sewage sludge did not have a significant effect on sulfur and its content was 0.06%, which is consistent with the literature [65,69].

The presence of oxygen in the fuel is certainly undesirable, since oxygen neither burns nor participates in combustion. It occurs only in compounds with other elements and thus decreases their combustibility as well as calorific value. The studied samples of Virginia mallow after the application of sewage sludge did not contribute to a significant change in oxygen content, and the mean value was 42.07%, which is still within the literature showing that biomass contains about 40% oxygen on average [70].

The structural composition of biomass is the optimal indicator of what type of biofuel a particular feedstock is best suited for. Lignin-rich biomass has greater potential in heat or power generation and direct combustion [63], while higher cellulose and hemicellulose content favors the production of liquid biofuels.

Table 7 shows that the content of cellulose was not significantly affected by sewage sludge, but hemicellulose and lignin were. It is interesting to note that hemicellulose content was increased by a higher percentage when treated with 1.66 t/ha of sewage sludge than when treated with 6.64 t/ha of sewage sludge application, from which it can be concluded that sewage sludge does not directly affect hemicellulose content. A similar situation was observed for lignin, only a decrease in the percentage of biomass. Thus, the lignin content at the 1.66 t/ha application was 16.51%, while the lignin content at the 3.32 t/ha treatment decreased to 15.91% and increased again to 16.77% at the 6.64 t/ha sewage sludge application.

**Table 7.** Structural composition of Virginia mallow biomass after application of different amounts of sewage sludge.

| Unit | % | | |
|---|---|---|---|
| **Sludge Treatment** | **Cellulose** | **Hemicellulose** | **Lignin** |
| T1 | 56.12 [a] | 19.70 [ab] | 17.97 [b] |
| T2 | 54.45 [a] | 21.17 [b] | 16.51 [a] |
| T3 | 56.46 [a] | 19.33 [a] | 15.91 [a] |
| T4 | 54.84 [a] | 20.20 [ab] | 16.77 [a] |
| Average | 55.47 | 20.10 | 16.79 |
| St.dv. | 1.49 | 1.28 | 0.88 |
| Significance | ns | *** | *** |

T1—(control), T2—1.66 t/ha, T3—3.32 t/ha, T4—6.64 t/ha. Values are the mean SD of three replicates. Different letters in each row indicate significant difference at $p \le 0.05$ by Tukey's HSD test. Values without letters are not significantly different at $p \le 0.05$ by Tukey's HSD test. Significance: *** $p < 0.001$; ns, non-significant.

The lignin content decreased with the increasing amount of sewage sludge, which could be a problem in direct incineration [63]. On the other hand, hemicellulose content increased with the amount of sewage sludge used, which is positive for liquid biofuel production. Virginia mallow has the best potential for use in bioethanol production due to its high cellulose content.

After the combustion process, an ash deposit remains that contains macro- and microelements, including heavy metals. In this study, the content of macronutrients in the biomass increased with the use of increasing amounts of sewage sludge, as shown in Table 8. K, Ca, Na, and Mg had the lowest content in the control fields, while after the application of the largest amount of sewage sludge, the content of the elements increased significantly.

**Table 8.** Macro-and microelement content of Virginia mallow biomass after application of different amounts of sewage sludge.

| Unit | mg/kg | | | | | | | | | | |
|---|---|---|---|---|---|---|---|---|---|---|---|
| Sludge Treatment | K | Ca | Mg | Na | Zn | Mn | Cr | Pb | Cu | Mo | Ni |
| T1 | 1626.72 [a] | 3075.94 [a] | 650.69 [a] | 23.66 [a] | 6.77 [a] | 23.66 [a] | 0.53 [a] | 0.74 [a] | 5.55 [d] | 0.62 [a] | 0.16 [a] |
| T2 | 1969.64 [b] | 3996.78 [c] | 787.86 [b] | 30.74 [b] | 8.93 [b] | 30.74 [b] | 0.69 [b] | 0.86 [b] | 1.80 [b] | 0.65 [a] | 0.16 [a] |
| T3 | 1654.33 [a] | 3362.67 [b] | 661.73 [a] | 25.87 [a] | 6.17 [a] | 25.87 [a] | 0.51 [a] | 0.70 [a] | 1.37 [a] | 0.60 [a] | 0.15 [a] |
| T4 | 2239.19 [c] | 4656.69 [d] | 895.68 [c] | 35.82 [c] | 10.74 [c] | 35.82 [c] | 0.83 [c] | 0.81 [b] | 2.34 [c] | 0.62 [a] | 0.16 [a] |
| Average | 1872.47 | 3773.02 | 748.99 | 29.02 | 8.15 | 29.02 | 0.64 | 0.78 | 2.77 | 0.63 | 0.16 |
| St.dv. | 378.83 | 563.63 | 151.53 | 4.34 | 1.58 | 4.34 | 0.14 | 0.07 | 1.81 | 0.08 | 0.02 |
| Significance | *** | *** | *** | *** | *** | *** | *** | *** | *** | ns | ns |

T1—(control), T2—1.66 t/ha, T3—3.32 t/ha, T4—6.64 t/ha. Values are mean ± SD of three replicates. Different letters in each row indicate significant difference at $p \leq 0.05$ by Tukey's HSD test. Significance: *** $p < 0.001$; ns, non-significant.

As for heavy metals, according to [71], Virgin mallow has the ability to grow well on marginal soils of poor quality, where it even contributes to soil stabilization. This is confirmed by the result of this study, where the Cu content was significantly reduced at the highest application rate of sewage sludge. In addition, [72] found Cu bioaccumulation when calcium-rich ash was applied to the soil. Copper is an undesirable component in biomass composition, and the result where its amount decreases is positive for biofuel production. Mo and Ni are two heavy metals that were not affected by the amount of sludge applied. A statistically significant difference was found for the elements Zn, Mn, Cr, P, and Cd when the maximum amount of sewage sludge was applied. Moreover, the amount of 3.32 t/ha resulted in similar control results for all these elements, so it is ungrateful to conclude that only the amount of sewage sludge applied influenced the increase in the amount of these metals.

However, the highest percentage of heavy metals collected was associated with the lowest dose applied of 1.66 t/ha of sewage sludge.

## 4. Conclusions

The rising price of mineral fertilizers on the market encourages the reuse of nutrients and organic residues from wastes such as sewage sludge. There is a growing demand for new, highly efficient, and low-cost energy crops that, growing on marginal soils, solve the problem of competition with food crops. In this study, the use of sewage sludge in the cultivation of Virginia mallow did not increase the content of heavy metals in the soil, which is potentially the biggest problem when sewage sludge is applied to the soil.

As for the energetic properties, the coke content in the highest dose applied was 11.19%, the ash content was 2.76%, and the upper heating value was 17.26 MJ/kg. The biomass yield also shows a positive trend when the applied amount of sewage sludge is increased. At an application rate of 6.64 t/ha, the yield was 8.85 t/ha. An increase was also

observed in the length of the plant itself, with plants averaging 3.12 m in length without treatment, while the application of 6.64 t/ha resulted in a plant height of 3.28 m.

The levels of micro- and macroelements and heavy metals in the biomass were optimal regardless of the amount of waste sludge applied during for growth of Virginia mallow. On the other hand, higher doses of municipal waste sludge contributed to the reduction in copper in the biomass. For example, untreated biomass was found to contain 5.55 mg/kg of copper, while biomass to which 6.64 t/ha of municipal sewage sludge was applied contained only 2.34 mg/kg of copper. Even better results were obtained for nickel and molybdenum, whose content in the biomass remained the same in the maximum applied amount of sewage sludge of 6.64 t/ha as in the untreated biomass. When considering the lignocellulosic composition, it was found that sewage sludge does not affect the cellulose content, while the hemicellulose content increases by 2.5% when 6.64 t/ha is applied and by 7.5% when 1.66 t/ha of sewage sludge is applied. It was concluded that the use of stabilized sewage sludge has a great potential for growing energy crops, which can lead to one of the most effective solutions for successful sewage sludge management. A potential problem is certainly the legislation, which has no clear guidelines for the use of sewage sludge for agricultural purposes at the level of the whole European Union. For this very reason, additional research on the use of sewage sludge as a soil amendment is needed to raise people's awareness, which could ultimately change the current view of sewage sludge as a worthless, unusable commodity.

**Author Contributions:** Conceptualization, J.Š., N.B., A.P., and I.B.; methodology, T.K., J.L., and M.P.; validation, J.Š., A.P., N.V., and N.B.; formal analysis, I.B. and A.P.; writing—original draft preparation, J.Š., N.B.; writing—review and editing, N.V., A.P, J.L., I.B., and T.K.; visualization, H.K. and J.L.; supervision, M.P., N.V., and N.B.; project administration, N.V. All authors have read and agreed to the published version of the manuscript.

**Funding:** This research was funded by the Croatian Science Foundation (HRZZ) under project No. IP-2018-01-7472, "Sludge management via energy crops' production", and within the project "Young Researchers' Career Development Project—Training of Doctoral Students", co-financed by the European Union, under the OP "Efficient Human Resources 2014–2020" from the ESF funds.

**Data Availability Statement:** Data generated during the study can be obtained by the authors of this study.

**Acknowledgments:** This research was cofounded by the Croatian Science Foundation (HRZZ) within the project "Young Researchers' Career Development Project—Training of Doctoral Students" (DOK-01-2018), co-financed by the European Union, under the OP "Efficient Human Resources 2014–2020" from the ESF funds.

**Conflicts of Interest:** The authors declare no conflicts of interest. The funders had no role in the design of the study; in the collection, analyses, or interpretation of data; in the writing of the manuscript; or in the decision to publish the results.

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
