# Peer review of "Wastewater Sewage Sludge Management via Production of the Energy Crop Virginia Mallow"

_agronomy, doi:10.3390/agronomy12071578_

Round 1

Reviewer 1 Report

The Manuscript “Wastewater sewage sludge management via production of the energy crop Virginia Mallow” requires revision before accepted for publication. The specific comments are given below.

1.     In the abstract, provide the most important numerical results of your research.

2.     The introduction should mention the effects of sludge treatment (composting, methane fermentation):

https://doi.org/10.3390/en14030590

https://doi.org/10.1016/j.biortech.2019.03.062

3.     In the last paragraph of the introduction, clearly indicate the research hypothesis.

4.     Ln 124 – Give the parameters of the work of the sewage treatment plant in Zagreb.

5.     Statistical analyzes are very important in research manuscripts. How were the normality of the distribution and the differences between the variables found?

6.     Please indicate the manufacturer, city, country when mentioning the equipment.

7.     Present the results obtained by other authors in a tabular form.

8.     Little cited literature from 2017-2022. Refresh references.

9.     Extend your conclusions with the most important results (numerical values).

Reviewer 2 Report

I have gone through the manuscript entitled "Wastewater sewage sludge management via production of the energy crop Virginia Mallow ". This study evaluated the influence of municipal sewage sludge on energy properties and biomass yield of Virginia mallow  The topic sounds interesting. The topic could belong the areas of interest to the Agronomy audience. The present manuscript could be considered to publish in the journal after a revision.

I would give the specific comments below that could help the authors improve their manuscript.

Abstract:

* The abstract should be revised. The authors should briefly discuss the purpose of the research and mention their findings adopted in this study. More quantitative data needs to be provided in the abstract.

* Provide significant words which are more relevant to the work in logical sequence as ‘keywords’. Also use keywords which are not present in title.

# Introduction:

* The introduction is incomplete and needs to be revised, reorganized, and rewritten. The main points that authors must discuss are (A) a brief discussion about the problem statement of the target problem in this study; (B) a brief discussion about the advantages/disadvantages of the applied methods for removing the target problem in the literature; then discuss the method assessed in this study and why it is suitable for the aim of this research; Why Virginia Mallow? (C) finally, (most importantly) discuss the novelty/originality of the present study and its difference with other researches published in this field.

* Tables; authors should provide the Tukeys test properly.  

*It would be necessary to develop more bioinformatic/statistical analyses in the present study. The information give is very scarce

# Results and discussion:

* Provide a better explanation for your data. Avoid only comparing your results to previous studies. Interpret and discuss the meaning of your results more deeply.  Discussions need to be supported by the latest references and need to be explained in depth. The authors should highlight the reason of their result findings in the light of available literature.

#Conclusions:
*Conclusions are mainly based on suppositions and not on the empirical results or literature evidence. All conclusions must be convincing statements on what was found to be novel, impactful based on strong support of the data/results/discussion. Limitations in the suggested approach should be discussed in the conclusions section. Numerical data should add in this section.

* Check and correct grammatical and space errors throughout the article.

Round 2

Reviewer 1 Report

Thank you for considering my comments.

Author Response

Dear reviewer,

Thank you for your valuable suggestions.

We have checked the spelling and also further controlled the English language and the style of the whole text.

Thank you for all your comments and suggestions, they were very helpful and certainly influenced the quality of our work.

Kind regards,

Authors

Reviewer 2 Report

The authors made all the corrections suggested in the paper. I recommend publishing.

Author Response

(The authors gave the same response as above.)
